# JAZF1, A Novel p400/TIP60/NuA4 Complex Member, Regulates H2A.Z Acetylation at Regulatory Regions

**DOI:** 10.3390/ijms22020678

**Published:** 2021-01-12

**Authors:** Tara Procida, Tobias Friedrich, Antonia P. M. Jack, Martina Peritore, Clemens Bönisch, H. Christian Eberl, Nadine Daus, Konstantin Kletenkov, Andrea Nist, Thorsten Stiewe, Tilman Borggrefe, Matthias Mann, Marek Bartkuhn, Sandra B. Hake

**Affiliations:** 1Institute for Genetics, Justus-Liebig University Giessen, 35392 Giessen, Germany; tara.procida@gen.bio.uni-giessen.de (T.P.); Tobias.Friedrich@biochemie.med.uni-giessen.de (T.F.); nadine.daus@gen.bio.uni-giessen.de (N.D.); Konstantin.kletenkov@gen.bio.uni-giessen.de (K.K.); 2Institute for Biochemistry, Justus-Liebig-University Giessen, 35392 Giessen, Germany; tilman.borggrefe@biochem.med.uni-giessen.de; 3Department of Molecular Biology, BioMedical Center (BMC), Ludwig-Maximilians-University Munich, 82152 Planegg-Martinsried, Germany; antoniajack2@gmail.com (A.P.M.J.); peritore@biochem.mpg.de (M.P.); clemensboenisch@gmail.com (C.B.); 4Department of Proteomics and Signal Transduction, Max Planck Institute of Biochemistry, 82152 Martinsried, Germany; hans-christian.h.eberl@gsk.com (H.C.E.); mmann@biochem.mpg.de (M.M.); 5Genomics Core Facility, Institute of Molecular Oncology, Member of the German Center for Lung Research (DZL), Philipps-University Marburg, 35043 Marburg, Germany; andrea.nist@imt.uni-marburg.de (A.N.); Thorsten.stiewe@uni-marburg.de (T.S.); 6Biomedical Informatics and Systems Medicine, Justus-Liebig-University Giessen, 35392 Giessen, Germany

**Keywords:** JAZF1, H2A.Z, histone variants, TIP60, acetylation, enhancer, gene regulation, ribosome

## Abstract

Histone variants differ in amino acid sequence, expression timing and genomic localization sites from canonical histones and convey unique functions to eukaryotic cells. Their tightly controlled spatial and temporal deposition into specific chromatin regions is accomplished by dedicated chaperone and/or remodeling complexes. While quantitatively identifying the chaperone complexes of many human H2A variants by using mass spectrometry, we also found additional members of the known H2A.Z chaperone complexes p400/TIP60/NuA4 and SRCAP. We discovered JAZF1, a nuclear/nucleolar protein, as a member of a p400 sub-complex containing MBTD1 but excluding ANP32E. Depletion of JAZF1 results in transcriptome changes that affect, among other pathways, ribosome biogenesis. To identify the underlying molecular mechanism contributing to JAZF1’s function in gene regulation, we performed genome-wide ChIP-seq analyses. Interestingly, depletion of JAZF1 leads to reduced H2A.Z acetylation levels at > 1000 regulatory sites without affecting H2A.Z nucleosome positioning. Since JAZF1 associates with the histone acetyltransferase TIP60, whose depletion causes a correlated H2A.Z deacetylation of several JAZF1-targeted enhancer regions, we speculate that JAZF1 acts as chromatin modulator by recruiting TIP60’s enzymatic activity. Altogether, this study uncovers JAZF1 as a member of a TIP60-containing p400 chaperone complex orchestrating H2A.Z acetylation at regulatory regions controlling the expression of genes, many of which are involved in ribosome biogenesis.

## 1. Introduction

Chromatin is the fundamental and functional building block of all eukaryotic genetic information localized in the cell’s nucleus [1]. It consists of DNA that is wrapped around an octamer of histone proteins—containing two of each of the histones H2A, H2B, H3 and H4—thereby forming a so-called nucleosome structure. Besides its ability to compact DNA, chromatin also has regulatory functions. It is, among others, involved in the regulation of gene expression, DNA damage repair, RNA processing, chromosome stability and cell cycle progression. Several interconnected mechanisms contribute to these functions, including chemical modifications of DNA itself or histone proteins, ATP-dependent nucleosome remodeling, non-coding regulatory RNAs and the deposition of histone variants [2]. The latter are specialized histones that differ in sequence, expression timing and function from their canonical counterparts [3]. Histone variants have been identified in all eukaryotes, and they contribute to distinct biological processes. Not surprisingly, deregulation of histone variants or factors that bind them and regulate their function and chromatin deposition can lead to diseases, most notably cancer [4].

One highly conserved histone variant of the H2A family is H2A.Z [5]; it differs from its canonical counterpart in 40% of its amino acid sequence. H2A.Z is essential in higher eukaryotes and plays an important role in all DNA-related processes, such as gene transcription and silencing, DNA damage repair, mitosis and development [6,7,8,9,10,11]. Upregulation of H2A.Z expression has been detected in several cancer types, e.g., prostate, colon, lung and skin [12,13,14,15,16].

In chordates, two non-allelic H2A.Z genes have evolved [17]. *H2AFZ* and *H2AFV* are located on separate chromosomes and differ greatly in their nucleotide sequences, both regulatory and coding. They give rise to two highly similar proteins, H2A.Z.1 and H2A.Z.2, respectively, that differ in only three amino acids from each other [18]. Previously, we were able to discover a novel primate-specific alternative splice variant of the H2A.Z.2 transcript, generating a C-terminally different histone protein, H2A.Z.2.2, which significantly destabilizes nucleosomes [19]. Regardless of this one functionally unique splice variant, H2A.Z.2.2, the question of how H2A.Z can play a role in so many different biological processes remains unresolved. In the last years, much progress has been made in identifying the proteins responsible for its chromatin deposition, as well as the genome-wide localization of H2A.Z-containing nucleosomes in many organisms as a first indication of its place of action. In higher eukaryotes, H2A.Z is deposited by the p400/TIP60/NuA4 (p400) and SRCAP complexes and is evicted by the INO80 complex or ANP32E in case of DNA damage [20,21,22]. In humans, H2A.Z is found at promoters, where it specifically localizes to two nucleosomes surrounding the nucleosome-depleted transcriptional start site (TSS), the so called −1 and +1 nucleosomes. H2A.Z is rather depleted from transcribed coding regions and has an inverse correlation with DNA methylation [23]. Furthermore, H2A.Z marks enhancer and silencer sequences, as well as euchromatin/heterochromatin borders that are characterized by the presence of the transcriptional repressor CCCTC-binding factor (CTCF) [24,25]. Moreover, several studies have identified H2A.Z as a constituent of pericentric heterochromatin [26] or even of centromeres [27], albeit these reports are somehow conflicting. Taken all of these studies into account, one picture emerges in which H2A.Z is present at regulatory genomic sites that control transcription, as well as cell cycle processes.

In this study, we identified and performed a site-by-site comparison of the human histone H2A variant chaperone complement that includes canonical H2A and the histone variants H2A.X, H2A.Bbd (also termed H2A.B) and H2A.Z.1. Interestingly, we found the transcriptional repressor and zinc finger protein JAZF1, together with MBTD1 and ANP32E, to be pulled-down with nuclear chromatin-free H2A.Z. Green Fluorescent Protein (GFP)-JAZF1 pull-downs, followed by quantitative mass spectrometry analyses, revealed that this zinc finger protein, together with MBTD1 but not ANP32E, is part of an H2A.Z-specific p400 chaperone/remodeling sub-complex. Global expression of many genes was found to be affected upon JAZF1 depletion, with several deregulated genes belonging to the ribosome biogenesis pathway. To gain insight into the molecular mechanism by which JAZF1, possibly together with the p400 complex, regulates transcription, we performed ChIP-seq experiments upon JAZF1 knockdown. Depletion of JAZF1 did not affect H2A.Z deposition, but led to a strong decrease of H2A.Z acetylation at more than 1000 regulatory regions, mostly located within introns. As knockdown of the MYST-family lysine acetyltransferase (KAT) TAT-Interactive Protein 60 (TIP60/KAT5/HTATIP) showed similar effects on H2A.Zac levels at JAZF1-targeted regulatory regions, we speculate that JAZF1 recruits this enzyme, together with the p400 complex to enhancers, to ensure a stable expression of target genes.

Taken together, our results provide interesting new insights into the functional histone H2A.Z variant network necessary for chromatin modifications and specific gene expression regulation.

## 2. Results

### 2.1. Identification of Chaperone/Remodeling Complexes of Human H2A Variants

Deposition of histone variants in a temporal- and spatial-restricted manner is crucial for the regulation of many DNA-based processes. In order to gain more insights into the requirements for proper histone exchange, we set out to identify the chaperone/remodeling complexes of almost all members of the human H2A variant family in a comparable and quantitative manner. We employed a quantitative mass spectrometry (MS)-based approach, similar to the one we previously used to successfully identify chaperone complexes of H2A.Z.2.1, H2A.Z.2.2 [19] and diverse human histone H3 variants [28]. Soluble nuclear proteins were isolated from stable isotope labeling by amino acids in cell culture (SILAC)-treated HeLa Kyoto cells expressing GFP, GFP–H2A, –H2A.X, –H2A.Z.1 or –H2A.Bbd. GFP-tagged canonical histone H2A or the respective GFP–H2A variants were precipitated and bound proteins identified by SILAC–MS (Figure 1).

As shown previously, GFP–H2A pulled-down, among others, both subunits of the FACT (facilitates chromatin transcription) complex [29,30,31] and NAP1L1 (nucleosome-assembly protein 1-like 1) [32] (Figure 1A), verifying the feasibility and accuracy of our approach. Additionally, we observed a strong interaction of nuclear soluble GFP–H2A with the XRCC5 and XRCC6 heterodimer of the Ku complex, which is implicated in DNA damage repair processes [33]. Further, we found GFP–H2A associating with MCM2, MCM4 and MCM6 (mini chromosome maintenance) proteins involved in DNA replication initiation and elongation [34], which, together with MCM7, have been previously found to also interact with H3.1, H3.2 and H3.3 histones [28,35]. Interestingly, H2A.X, a histone variant involved in DNA damage repair [36,37], and differing only in the C-terminal tail from H2A, also interacted with NAP1L1, the FACT [38] and Ku complexes, as well as the three MCM proteins (Figure 1B). It is therefore likely that both histone proteins share the same chaperone recognition site(s) and are deposited into similar genomic location. GFP–H2A.Bbd, on the other hand, only bound the FACT but not the Ku complex, nor MCM proteins (Figure 1C). These findings are in line with the observation that H2A.Bbd might be transiently deposited into chromatin during replication, when the chromatin is open and accessible [39], and possibly through an active mechanism during transcription via FACT [31]. Similar to the other GFP–H2A variants, GFP–H2A.Z.1 was also found in a complex with the FACT and Ku complexes, but not MCM proteins (Figure 1D), suggesting a general recognition mechanism between these factors and histone H2A variants. In agreement with our previous data on H2A.Z.2.1 and H2A.Z.2.2 chaperone complexes [40], we were able to identify all members of the p400/TIP60/NuA4 (p400) and SRCAP complexes bound to GFP–H2A.Z.1, as well as ANP32E, which has been shown to eject H2A.Z from nucleosomes [22,41,42,43] (Figure 1D). Interestingly, in addition to other proteins, we also observed the interaction of GFP–H2A.Z.1, as well as GFP–H2A.Z.2.1 and –H2A.Z.2.2 [40], with the transcriptional repressor Juxtaposed with Another Zinc Finger 1 (JAZF1; formerly named TIP27) [44] and the MBT-domain-containing protein MBTD1 [45], which has previously been show to interact with the p400 complex and to be involved in DNA damage repair regulation [46].

In summary, all H2A variants are recognized by the FACT complex; H2A, H2A.X H2A.Z are bound by the Ku complex; H2A and H2A.X recognize MCM proteins; and H2A.Z has a special status in binding two large chaperone complexes, namely p400 and SRCAP (Figure 1E).

### 2.2. JAZF1 Is a Member of an MBTD1-Containing But ANP32E-Excluding H2A.Z-Specific p400 Chaperone Sub-Complex

Having identified JAZF1 as novel H2A.Z-associated protein, we wondered whether it directly interacts with this histone variant alone or as member of the p400 and/or SRCAP chaperone complexes. We therefore performed SILAC–MS with soluble nuclear extracts from HeLa Kyoto cells transiently expressing GFP–JAZF1. Interestingly, GFP–JAZF1 associated with all members of the p400 and shared p400/SRCAP complexes, but did not pull-down any specific and unique SRCAP complex proteins, such as SRCAP, ZNHIT1 or ACTR6 (Figure 2A,B).

Additionally, GFP–JAZF1 pulled-down MBTD1, suggesting that both proteins are part of one p400 complex (Figure 2A,B). Surprisingly, GFP–JAZF1 pull-downs did not contain ANP32E, which has previously been demonstrated to also be part of the p400 complex and to remove H2A.Z from chromatin [22,41]. To verify binding of JAZF1 to the p400 complex, we performed GFP and GFP–TIP60, the lysine acetyltransferase (KAT) member of the p400 complex, pull-down experiments followed by immunoblots with antibodies against JAZF1 or, as negative control, the SRCAP-specific ZNHIT1 protein, and showed that GFP–TIP60 indeed interacts with endogenous JAZF1 (Figure 2C). Unfortunately, the reciprocal experiment pulling-down GFP–JAZF1 was not applicable, as commercially available antibodies against TIP60 cross-reacted with GFP–JAZF1.

In summary, we identified JAZF1 as a novel member of an ANP32E-independent p400 chaperone sub-complex that contains MBTD1.

### 2.3. JAZF1 Depletion Leads to Deregulation of Genes Involved in Ribosome Biogenesis

Next, we investigated JAZF1’s possible role in regulating gene expression, as this protein was previously shown to act as transcriptional repressor of the nuclear orphan receptor TAK1/TR4 [44], and because the associated p400 complex has also been implicated in the regulation of transcription [47]. We successfully knocked-down JAZF1 (Appendix A), using a JAZF1-specific siRNA pool in HeLa Kyoto cells. RNA-seq analyses of JAZF1 depleted cells from two time-distinct batch experiments containing two or three respective biological replicates identified 162 genes to be significantly deregulated (Figure 3A and Appendix A), a result verified by RT-qPCR (Figure 3B).

Gene set enrichment analysis (GSEA) based on the differential expression (siJAZF1 vs. siNTC) of all genes revealed a statistically significant enrichment of factors involved in ribosome biogenesis, among other pathways (Figure 3C and Appendix A). This observation is in line with the finding that JAZF1 localizes not only to the nucleus, but also that observed bright JAZF1 speckles overlap with some nucleoli, as immunofluorescence (IF) microscopy pictures of co-staining of endogenous JAZF1 with Fibrillarin, a marker for nucleoli, demonstrate (Figure 3D, top). We also noticed that the nucleoli sizes increased upon JAZF1 knockdown. These observations are in agreement with a previous report linking JAZF1 with ribosomal protein synthesis in pancreatic β-cells and larger nucleoli sizes upon JAZF1 depletion [48]. Interestingly, many JAZF1 speckles also co-localize with Coilin (Figure 3D, bottom), a marker of Cajal bodies [49], and are found in close proximity to the nucleolus and implicated in the modification of non-coding RNAs, including those forming the core of small nucleolar ribonucleoproteins (snoRNPs) required for ribosomal RNA (rRNA) processing [50].

Taken together, JAZF1 is a nuclear/nucleolar/Cajal-body protein involved in the regulation of expression of many genes, with several of them involved in ribosome biogenesis.

### 2.4. JAZF1 Depletion Does Not Influence H2A.Z Occupancy But Leads to Reduced H2A.Z Acetylation Levels at Intronic Enhancer Sites

Our data clearly demonstrate that JAZF1 is important for controlling gene regulation and that it interacts with the TIP60-containing p400 complex, which has been implicated in the deposition of H2A.Z [51,52] and acetylation of, among other histones, H2A.Z [53]. We speculate that JAZF1 might be able to recruit the enzymatically active p400 complex to distinct sites in the genome, where it might affect H2A.Z levels, its localization sites and/or its acetylation, thereby impacting gene expression of some target genes. While we were not able to determine JAZF1’s genome localization, as several antibodies proved to be not suitable for ChIP-seq assays, we set out to evaluate the influence of JAZF1 on H2A.Z within chromatin. First, we evaluated a possible effect of JAZF1 on global histone acetylation levels, as we found it bound to the KAT TIP60 (see Figure 2). However, upon JAZF1 depletion (Appendix A), we detected no global changes in the level of H2A.Z or other histone acetylation sites by immunoblots (Appendix A). Next, we performed ChIP-seq experiments with antibodies against H2A.Z, to identify possible changes in its chromatin deposition, or with antibodies against H2A.Z acetylation (H2A.Zac), to analyze a possible effect on TIP60-mediated modification on specific genomic regions. First, comparing JAZF1 with control knockdown H2A.Z ChIP-seq peaks, we observed no changes in the level and localization of H2A.Z at promoters and enhancers [54,55] (Appendix A), an observation we were also able to verify by ChIP–qPCR (Appendix A). This result suggests that JAZF1 does not influence chromatin deposition or ejection mechanisms of H2A.Z, at least at those sites that are accessible for bioinformatics analyses. However, we identified >1000 sites showing significant changes in H2A.Z acetylation levels (Figure 4A,B).

Strikingly, JAZF1 depletion resulted in an overall reduction in the level of H2A.Zac at those particular genomic regions (Figure 4C). In order to determine the identity of these differentially acetylated H2A.Z sites (diffH2A.Zac) upon JAZF1 depletion, we compared those sites with chromatin states defined by training a 10-state model on ENCODE data for H3K4me3, H3K4me1, H3K27ac, H3K36me3 and H3K27me3, using ChromHMM [56,57] (Figure 4D). Sites with a significant decrease of H2A.Zac signal were observed at active enhancers within gene bodies (states 2 and 6), which are marked by H3K36me3, H3K4me1 and H3K27ac, but not at active promoters that are characterized by high H3K4me3 levels (states 1 and 3) (Figure 4E). More specifically, these differentially acetylated H2A.Z sites are enriched within introns of genes (Appendix A) and could be independently confirmed by ChIP–qPCR (Figure 4F). In order to test whether TIP60, which we found to be associated with JAZF1 (see Figure 2), is responsible for H2A.Z acetylation at those particular regulatory sites, we knocked-down this KAT (Appendix A) and demonstrated that TIP60 depletion also results in an decrease in H2A.Zac at JAZF1-dependent differentially modified enhancer sites (Figure 4F).

In conclusion, JAZF1 is crucial for the acetylation of H2A.Z at some regulatory chromatin regions within introns, possibly by recruitment of a histone acetyltransferase, such as the p400 complex member TIP60.

## 3. Discussion

In this work, we have identified potential chaperone/remodeling complexes of many human histone H2A variants, with the exception of macroH2A, which has previously been shown to be deposited into chromatin via a transcription-associated “pruning” mechanism involving FACT [58]. We found NAP1L1 binding to H2A and H2A.X, and this strengthens the feasibility and specificity of our approach, as it was previously shown by several groups to facilitate H2A chromatin assembly [59,60]. Interestingly, in line with other reports [38,39], we observed that all H2A variants are associated with the FACT complex, probably via H2B, suggesting that these histones are incorporated (also) in a transcription- and or replication-dependent manner when polymerases modulate the underlying chromatin structure [31]. Interestingly, all H2A variants, with the exception of H2A.Bbd, did interact with the Ku complex that is involved in DNA damage repair [33]. Since H2A.Bbd is highly divergent in its amino acid sequence, as compared to the other H2A variants, it is plausible to speculate that the observed interaction is direct between one conserved H2A variant protein sequence and one of the Ku complex proteins and not indirectly mediated via H2B. The observed interaction of specifically H2A and H2A.X with MCM proteins suggests a role of these proteins in the replication-dependent chromatin incorporation of these variants. As H2A.X is expressed at a lower level than canonical H2A, one can envision that, during replication, a high level of H2A and less of H2A.X proteins are deposited during replication, ensuring a faithful packaging of the underlying DNA in combination with a regular and random distribution of this important DNA damage marker protein H2A.X. In addition, we also identified other proteins associated with some H2A variants. Mesoderm induction early response 1 (MIER1) [61] and AT-rich interaction domain 5B (ARID5B) [62] proteins were enriched after H2A, H2A.X and H2A.Z pull-downs. However, their putative role in histone deposition needs to be evaluated in future studies.

Importantly, we have identified JAZF1 as member of the p400 complex in HeLa Kyoto cells, thereby confirming that such a complex can exist in different cell types and differentiation states, as this interaction was previously only found in human endometrial stromal cells [63]. Our identification of JAZF1 with MBTD1, a chromatin-binding protein shown in an earlier study to pull-down the p400 complex [46], is an exciting discovery and opens up several questions for future research. Do JAZF1 and MBTD1 interact with each other? How are they binding to the p400 complex, and what factor within this complex do they recognize? As the presence of both MBTD1 and JAZF1 with p400 leads to a likely exclusion of ANP32E, how is this achieved? Do they compete with the same binding site? Moreover, it will be of interest to evaluate how the DNA damage repair function of MBTD1 [46] links to the regulation of gene expression, as observed for JAZF1 in our and other studies [44].

In HeLa Kyoto cells, we found JAZF1 to be important for the regulation of several genes, many of them involved in ribosome biogenesis. This observation is in agreement with a publication by Kobiita et al. showing JAZF1 to be a principle transcriptional regulator of ribosome biogenesis in pancreatic β-cells [48] and suggests that JAZF1’s regulatory function is not cell-type specific. Further, we were able to show that JAZF1 is crucial for the acetylation of the histone variant H2A.Z at more than 1000 regulatory regions, mostly found within introns. It is highly probable that TIP60 enzymatically catalyzes this modification, as JAZF1 and TIP60 are found within one complex, and TIP60 depletion leads to decreased H2A.Zac levels at some JAZF1-affected sites. We speculate that JAZF1 recruits TIP60 to some intronic regulatory regions, where it acetylates close-by H2A.Z-containing nucleosomes, to ensure a tightly controlled transcriptional output of their target genes. Some of those might be directly or, more likely, indirectly involved in the regulation of ribosome biogenesis factors. Support for this hypothesis comes from the study by Halkidou et al. which demonstrates a role for TIP60 in rDNA and ribosomal gene transcription in prostate cancer cell lines [64]. Since we observed similar effects in HeLa Kyoto cells, it is likely that JAZF1, together with TIP60, is a general and cell-type-independent regulator of ribosome pathways. Nevertheless, since deregulation of JAZF1 expression has been strongly correlated with type 2 diabetes (T2D) and oncogenesis [65,66,67,68,69] in many studies, it is probable that the transcriptional regulatory function of this zinc finger protein is of particular importance in islet or cancer cells.

One remarkable finding is the enhanced frequency of H2A.Zac-reduced sites within intronic enhancers upon JAZF1 depletion. We were not able to identify any particular DNA motifs to be enriched within these genomic sites, arguing against a particular DNA sequence responsible for direct recruitment of TIP60 via JAZF1. Although JAZF1 has been linked to E26 transformation-specific (ETS) and direct repeat DNA-element (DR1) transcription factor motifs within transcriptional start sites (TSS) [48], we only found changes in H2A.Z acetylation levels in enhancer but not promoter regions. One possible explanation might be differential and/or cell-type specific chromatin/DNA recruitment mechanisms of JAZF1 and/or the occurrence of distinct JAZF1 complexes. We were particularly interested in JAZF1’s connection with H2A.Z and its chaperone complex p400 and therefore looked specifically at associated chromatin changes, while other studies investigated HA-tagged JAZF1 DNA binding sites by ChIP-seq. It will be interesting to determine, in the future, whether JAZF1 can act alone or might be a member of several different complexes directly on chromatin, since, in this study, we only looked at JAZF1’s chromatin-free binding partners.

In summary, our study reveals JAZF1 as member of an MBTD1-containing and ANP32E-excluding p400 chaperone/remodeling complex with a functional role in H2A.Z acetylation at regulatory regions.

## 4. Materials and Methods

### 4.1. Cell Culture, Transfection and Flow Cytometry

The cervical cancer cell line HeLa Kyoto, a gift from Heinrich Leonhardt (LMU Munich, Munich, Germany), was grown in Dulbecco’s Modified Eagle’s Medium (DMEM, Thermo Fisher Scientific, Waltham, MA, USA) supplemented with 10% heat-inactivated fetal bovine serum (FBS, Thermo Fisher Scientific) and 1% penicillin/streptomycin (Thermo Fisher Scientific), in a humidified atmosphere, at 37 °C and 5% CO_2_. HeLa Kyoto cells stably expressing enhanced green fluorescence protein (GFP), GFP-tagged H2A or GFP-tagged histone variants (H2A.X, H2A.Bbd and H2A.Z.1) were cultured as described [16,19]. Transient transfections of HeLa Kyoto cells with GFP, GFP–JAZF1 or GFP–Tip60 were performed by using X-tremeGENE HP (Roche, Basel, Switzerland) or FuGENE HD (Promega, Madison, WI, USA) according to the manufacturer’s instructions. Two days after transfection, cells were harvested for several experimental applications. Expression levels of stably or transiently transfected HeLa Kyoto cells were quantified by using a FACSCanto machine (Becton Dickinson, Franklin Lakes, NJ, USA) or a BD Accuri C6 Plus Flow Cytometer (Becton Dickinson). Cells were routinely tested for mycoplasma contamination.

### 4.2. Plasmids

The cloning of GFP und GFP–H2A variant expression plasmids is described in References [19,39]. GFP–JAZF1 (GFP–TIP27) plasmid was a gift from Anton Jetten [44], and GFP-TIP60 plasmid was a gift from Benoit de Crombrugghe [70].

### 4.3. SILAC Followed by Mass Spectrometric (MS) Identification

HeLa Kyoto cells stably expressing GFP, GFP–H2A, GFP–H2A.X, GFP–H2A.Bbd and GFP–H2A.Z.1 and HeLa Kyoto cells transiently expressing GFP or GFP–JAZF1 were SILAC (stable isotope labeling, using amino acids in cell culture) labeled, chromatin-free nuclear cell extracts prepared and mass spectrometric experimental procedure was performed as described [19,28,71,72]. Lists of all identified proteins are shown in Appendix A. Scatter plot and heatmap analyses were performed by using R language for statistical computing [73], RStudio [74] and “ggplot2” package [75].

### 4.4. Antibodies

The following primary antibodies were used for Immunoblotting, ChIP or IF applications: α-JAZF1 (1:500, produced by Pineda, animal 1), α-ZNHIT1 (1:1000, HPA019043, Sigma-Aldrich, Soeborg, Denmark), α-GFP (1:1000, 11814460001, Sigma-Aldrich), α-H2A.Z (1:1000 or 5 µg/IP, 39944, Active motif, Carlsbad, CA, USA), α-H2A.Zac (1:1000 or 1 µg/IP, ABE1363, Merck Millipore, Darmstadt, Germany), α-alpha Tubulin (1:1000, 39527, Active motif), α-H4K5ac (1:1000, 39700, Active motif), α-H4K16ac (1:1000, 39167, Active motif), α-H2AK5ac (1:1000, 39108, Active motif), α-H3K14ac (1:1000, 39599, Active motif), α-H3 (1:1000, ab1791, Abcam, Cambridge, MA, USA), α-JAZF1 (1:100, HPA066967, Atlas Antibodies, Stockholm, Sweden), α-Fibrillarin (1:100, NB300-269, Novus Biologicals, Centennial, CO, USA), α-Coilin (gift from Grahmam Dellaire). The following secondary antibodies were used for immunoblotting or IF applications: α-rabbit-HRP (1:20000, 31460, Thermo Fisher Scientific), α-mouse-HRP (1:20000, 31430, Thermo Fisher Scientific), α-mouse-Alexa Fluor 488 (1:200, A-11017, Thermo Fisher Scientific), α-rabbit-Alexa Fluor 594 (1:200, A-11072, Thermo Fisher Scientific), α-rabbit-Alexa Fluor 488 (1:200, A-11070, Thermo Fisher Scientific), α-mouse-Alexa Fluor 594 (1:200, A-11020, Thermo Fisher Scientific).

### 4.5. Immunoprecipitation of GFP-Tagged Proteins

HeLa Kyoto cells transiently expressing GFP or GFP–TIP60 and non-transfected HeLa Kyoto cells were used for the preparation of nuclear extracts. Cells were resuspended in 200 µL ice-cold RIPA buffer (10 mM Tris/Cl pH 7.5, 150 mM NaCl, 0.5 mM EDTA, 0.1% SDS, 1% Triton X-100, 1% deoxycholate) supplemented with DNase I (75 Kunitz U/mL), MgCl_2_ (2.5 mM), dithiothreitol (DTT, 1 mM) and protease inhibitors. After incubation on ice for 30 min and centrifugation of cell lysates at 17,000× *g* for 10 min at 4 °C, the supernatant was diluted with 300 µL Dilution buffer (10 mM Tris/Cl pH 7.5, 150 mM NaCl, 0.5 mM EDTA) supplemented with DTT and protease inhibitors. For input fraction, 50 µL (10%) of diluted lysates was saved. For immunoprecipitation, 20 µL of GFP–Trap Dynabeads (Chromotek, Munich, Germany) was washed once with 500 µL ice-cold Wash buffer (Dilution buffer containing 0.05% Nonidet P40 Substitute). After magnetic separation of beads, the supernatant was discarded, and diluted lysates were added to the equilibrated beads and incubated over night (ON) at 4 °C (end-over-end rotation). The following day, samples were washed three times with 500 µL Wash buffer supplemented with DTT and protease inhibitors. Elution of immunoprecipitated proteins was achieved by boiling beads in 2× Laemmli sample buffer. Separation and detection of proteins was performed by sodium dodecyl sulfate polyacrylamide gel electrophoresis (SDS-PAGE), followed by immunoblotting.

### 4.6. Immunoblotting

Whole-cell extracts were achieved by lysing cells in RIPA buffer, followed by the addition of Laemmli sample buffer and boiling of cells. Lysates were subjected to SDS-PAGE, and gel-separated proteins were blotted onto a nitrocellulose membrane. For protein detection, the membrane was blocked in PBS-T containing 5% non-fat dry milk for 1 h, at room temperature (RT), followed by ON incubation at +4 °C with the indicated antibodies diluted in Blocking buffer. The following day, the membrane was washed three times with PBS-T and incubated with HRP-conjugated secondary antibodies diluted in Blocking buffer for 1 h, at RT. HRP signals were detected by using Trident femto Western HRP Substrate (GeneTex, Irvine, CA, USA).

### 4.7. siRNA Transfections

All ON-TARGETplus SMARTpool siRNAs (JAZF1 and Non-targeting control) were synthesized by Dharmacon (Lafayette, CO, USA). For JAZF1 gene silencing, the following pool of four different siRNAs (siJAZF1) was used: 5′-GGCAUAAAGUAUCACGCUA-3′, 5′-ACAGAUUCGUGUCCGCAAA-3′, 5′-GAUACAGAUCCACGGGUUU-3′ and 5′-GAUCCAGACAUGAGACGCA-3′. The non-targeting pool (siNTC) contains the following four siRNAs: 5′-UGGUUUACAUGUCGACUAA-3′, 5′-UGGUUUACAUGUUGUGUGA-3′, 5′-UGGUUUACAUGUUUUCUGA-3′ and 5′-UGGUUUACAUGUUUUCCUA-3′. TIP60-specific siRNAs were designed, synthesized and pre-validated as described previously in Reference [55]. The following double-stranded siRNAs were used: siTIP60 #1, 5′-GGAGAAAGAAUCAACGGA-3′; siTIP60 #2, 5′-CAACAAACGUCUGGAUGA-3′. For transient depletion of JAZF1 or TIP60, HeLa Kyoto cells were transfected with indicated siRNAs, using oligofectamine (Invitrogen, Carlsbad, CA, USA) according to the manufacturer’s instructions. Two days after transfection, cells were collected for several experimental applications. Knockdown efficiency was controlled by a combination of immunoblotting and RT-qPCR.

### 4.8. RNA Isolation, cDNA Preparation and RT-qPCR

HeLa Kyoto cells transfected with control (NTC), JAZF1 or TIP60 siRNAs were harvested, total RNA was isolated by utilizing the RNeasy Mini Kit (QIAGEN, Montreal, QC, Canada) and on-column DNase digestion (RNase-Free DNase Set, QIAGEN) was performed according to the manufacturer’s instructions. Quality control of RNA was achieved by electrophoresis on a 1.5% agarose gel. For reverse transcription, 1 µg of total RNA was reverse transcribed, using the Transcriptor First Strand cDNA Synthesis Kit (Roche, Basel, Switzerland), following the manufacturer’s protocol. Quantitative polymerase chain reaction (qPCR) of cDNA was performed in triplicates, using SYBR Green master mix (Bio-Rad, Copenhagen, Denmark) and a CFX96 Real-Time System (Bio-Rad) with the following settings: 5 min at 95.0 °C, followed by 40 cycles at 95.0 °C for 3 s and 60.0 °C for 20 s. The amount of PCR product was measured after each cycle, and a melting curve was generated (65.0 to 95 °C, plate read every 1.0 °C) to ensure the specificity of PCR product and primer pairs. PCR efficiency was determined by preparing a standard curve of serially diluted cDNA for each primer pair used in this study. Fold change gene expression levels were calculated based on the 2−ΔΔCT method relative to the siNTC negative control and normalized to the transcript level of the housekeeping gene HPRT1 (hypoxanthine phosphoribosyltransferase 1). Error bars are representing the standard deviation of technical triplicates. The following primer sets were used in this study:

HPRT1:

F: 5′-AAGGGTGTTTATTCCTCATGGA-3′

R: 5′-AATCCAGCAGGTCAGCAAAG-3′

JAZF1_1:

F: 5′-CAGATTCGTGTCCGCAAACC-3′

R: 5′-AATTGATTGTGTGGTGCCGC-3′

JAZF1_2:

F: 5′-CCGGTGTCGGCTGAGATTAT-3′

R: 5′-AGCAACTGCTGGTGAGGATT-3′

TIP60_1:

F: 5′-GAAGATGGCGGAGGTGGTG-3′

R: 5′-CGGCAGCCCTCGATTATCTC-3′

TIP60_2:

F: 5′-AGGGGGAGATAATCGAGGGC-3′

R: 5′-CTTCACGCTCAGGATCTCGG-3′

CAPN2:

F: 5′-CCCTAAACCAGAGCTTCCAGG-3′

R: 5′-CATCCACCACCACCTCCAC-3′

BIRC3:

F: 5′-TCAGACAGCCCAGGAGATGA-3′

R: 5′-CACGGCAGCATTAATCACAGG-3′

ITGB8:

F: 5′-AGAAGGAGGTTTTGACGCCAT-3′

R: 5′-TGTCATCACCAGCAGCAATCT-3′

LSM7:

F: 5′-GGACGGCACCATTGAGTACA-3′

R: 5′-GAAGGGGTTGGGGATGGC-3′

### 4.9. RNA-Seq and Data Analysis

Total RNA was isolated from control (NTC) and JAZF1 depleted HeLa Kyoto cells as described above. Quality control of RNA was achieved by a combination of agarose gel and capillary electrophoresis (Fragment Analyzer, Agilent, Santa Clara, CA, USA), using the RNA Kit (Agilent). Library preparation of total RNA, using the QuantSeq 3′ mRNA-Seq Library Prep Kit FWD (Lexogen, Vienna, Austria) and next-generation sequencing (75 bp read length, single-read), was performed on the Illumina NextSeq550 platform by the Laboratory of the Genomics Core Facility in Marburg (Philipps-University, Marburg, Germany).

RNA-seq data analysis was performed by using a customized systemPipeR R/BioConductor package [76,77] within R version 4.0.2 [73]. Random UMI (Unique Molecular Identifier) barcodes were extracted, copied to the corresponding header line and removed from the raw sequencing reads, using UMI-Tools [78] with –bc-pattern = NNNNNNXXXX parameter. These processed FASTQ files were quality and adaptor trimmed, and the 4 bases that belong to the specific UMI barcode were removed with Trim Galore v.0.6.5 [79], using –clip_R1 4 parameter. The resulting sequencing reads were aligned to the human genome (hg19) and the associated GTF (Gene Transfer Format) file (Illumina’s IGenomes) and stored as BAM files, using Hisat2 v.2.2.1 (Ref5) with –k 1 –min-intron 30–max-intron 3000 parameters. PCR duplicates were removed from these BAM files, using UMI-Tools [78], and a gene-annotation-counts table, including all samples, was calculated by using the summarizedOverlaps function of the GenomicAlignimnets R/BioConductor package [80]. Normalization and subsequent detection of differentially expressed genes was done by using DESeq2 v.1.28.1 [81] with two batches and both conditions (siJAZF1 and siNTC) as the chosen design variables. The volcano plot was generated by using the EnhancedVolcano R/BioConductor package [82] with the significant criteria log2(FC) > −0.5 and <0.5 and False Discovery Rate < 0.1. GSEA, based on the Wald statistic provided by DESeq2, was performed by using the clusterProfiler R/BioConductor package [83] with GO (Gene Ontology) and KEGG as databases. The universe for the GSEA was defined as all genes have read counts in at least one sample. Pathways were counted as statistically significant with an adjusted *p*-value < 0.1. Analysis code is available on request. Both, raw and processed data are available in the NCBI Gene Expression Omnibus (GEO), under accession number GSE163214.

### 4.10. Immunofluorescence (IF) Microscopy

IF stainings were performed as described in Reference [84], with a few adjustments. Briefly, HeLa Kyoto cells were cultured on coverslips and fixed with 3% paraformaldehyde in PBS. After three PBS washes, cells were permeabilized with PBS containing 0.5% Triton X-100, followed by blocking with 1% bovine serum albumin (BSA) in PBS. For target protein detection, coverslips were stepwise incubated with primary and then secondary Alexa Fluor–conjugated antibody. After washing, DNA was visualized by using 10 µg/mL Hoechst in PBS. Coverslips were mounted in Fluoromount-G mounting medium (SouthernBiotech, Birmingham, AL, USA). Images were acquired with an Axio Observer.Z1 inverted microscope (Carl Zeiss, Oberkochen, Germany) with Axiocam 506 mono system. Image processing was performed with Zeiss Zen 3.1 (blue edition) software and ImageJ (version 1.51n).

### 4.11. Histone Extraction

HeLa Kyoto cells exposed to control (NTC) or JAZF1 siRNA pools were harvested and collected at a density of 1 × 10^7^ cells. Acid extraction of histones was performed as described in References [85,86]. Separation and detection of histone proteins was performed by a combination of SDS-PAGE gel electrophoresis, followed by Coomassie Brilliant Blue staining and immunoblotting.

### 4.12. Chromatin Immunoprecipitation Followed by Sequencing (ChIP-Seq)

#### 4.12.1. Chromatin Immunoprecipitation (ChIP)

JAZF1-depleted or control (siNTC-transfected) HeLa Kyoto cells were fixed with DMEM medium containing 1% formaldehyde at a density of 1 × 10^7^ cells, for 10 min, at RT. After quenching with 125 mM glycine, the cells were washed three times with PBS. Fixed cells were resuspended in 1 mL SDS-Lysis buffer (50 mM Tris/Cl pH 8.1, 10 mM EDTA, 1% SDS) supplemented with DTT (1 mM) and protease inhibitors and then transferred to 15 mL conical hard plastic tubes. For chromatin shearing, cell suspension was subjected to Bioruptor sonication (15 cycles high energy, 30 s on–30 s off; Diagenode, Toyama, Japan) to generate chromatin fragments of 250 bp in average size. Sheared chromatin was centrifuged for 10 min, at 18,400× *g*, at 4 °C, and supernatant was used for immunoprecipitations (IPs). For input fraction, 10 µL (10%) of lysates was saved. The day before, 10 µL of magnetic Dynabeads Protein G (Invitrogen) was washed one time with Dilution buffer mix (90% Dilution buffer (16.7 mM Tris/Cl pH 8.1, 167 mM NaCl, 1.2 mM EDTA, 1% Triton X-100, 0.01% SDS and 10% SDS–Lysis buffer) and incubated with the indicated antibodies, ON, at 4 °C (end-over-end rotation). The following day, antibody-coupled Dynabeads were magnetic separated and washed three times with Dilution buffer mix. Immunoprecipitations were carried out by re-suspending beads in 900 µL Dilution buffer supplemented with DTT (1 mM) and protease inhibitors and by adding 100 µL of chromatin, followed by ON incubation at 4 °C, while rotating. The next day, beads were collected and washed 1x with Low-Salt buffer (20 mM Tris/Cl pH 8.1, 150 mM NaCl, 2 mM EDTA, 1% Triton X-100, 0.1% SDS), 1x with High-Salt buffer (Low-Salt buffer with 500 mM NaCl), 1× LiCl buffer (10 mM Tris/Cl pH 8.1, 1 mM EDTA, 0.25 M LiCl, 1% Deoxycholate, 1% NP40) and 2× with TE (Tris-EDTA) buffer (10 mM Tris/Cl pH 8.1, 1 mM EDTA). Afterwards, beads and input samples were resuspended in 100 µL TE buffer, and after the addition of 1 µL RNase A (10 mg/mL), the samples were incubated at 37 °C for 30 min. Next, 5 µL of 10% SDS and 2.5 µL of Proteinase K (20 mg/mL) were added to each reaction and incubated at 37 °C, for 4 h, and then switched to 65 °C ON incubation. DNA was purified by using the illustra GFX PCR DNA and Gel Band Purification Kit (GE Healthcare Life Sciences) according to the manufacturer’s instructions and eluted in 20 µL super-clean water. DNA concentrations were determined with a Qubit 4 Fluorometer (Invitrogen) and dsDNA HS Assay Kit (Invitrogen). For Illumina Sequencing, libraries were generated with the MicroPlex Library Preparation Kit (Diagenode), following the manufacturer’s protocol. Libraries were eluted in 20 µL 0.1× TE buffer pH 8.0 and quantified with a Fragment Analyzer (Agilent), using the HS Small Fragment Kit (Agilent).

#### 4.12.2. Illumina Sequencing

Next-generation sequencing (75 bp read length, single-read) was carried out on the Illumina NextSeq550 platform by the Laboratory of the Genomics Core Facility in Marburg (Philipps-University, Marburg, Germany). Both raw and processed data are available in the NCBI Gene Expression Omnibus (GEO), under accession number GSE163318.

#### 4.12.3. ChIP-Seq Analysis

FASTQ files were controlled for quality issues, using FastQC (https://www.bioinformatics.babraham.ac.uk/projects/fastqc/). Read alignment against the hg19 human genome reference was downloaded as a pre-compiled BWT index from Illumina’s iGenome repository (https://emea.support.illumina.com/sequencing/, sequencing_software/igenome.html). Read alignment was performed by using bowtie version 1.1.2 with parameters -k 1 -m 1. Duplicate removal was performed by using Picard’s MarkDuplicates and Samtools rmdup function. Coverage vectors were generated with Deeptools bamCoverage function, using RPKM (reads per kilo base per million mapped reads) normalization [87]. Visualization of binding profiles was done by using the R/BioConductor package Gviz. H2A.Z and H2A.Zac peak calling was done using MACS2 [88]. The resulting set was filtered against blacklisted chromatin regions, as detected by ENCODE. Differential binding analysis was performed by using DESeq2 [81] after merge peaks into reference peak sets, using reduce function. In order to identify overrepresented functional terms amongst binding-site-associated genes, we used GREAT (Genomic Regions Enrichment of Annotations Tool) [89], using standard settings. For heatmap representation of binding data across binding sites, coverage vectors were collected in defined intervals across binding sites or RefsSeq TSSs, using Deeptools computeMatrix, and plotted via Deeptools PlotHeatmap function. The comparison to chromatin states was performed as previously described [55].

### 4.13. ChIP–qPCR

After chromatin immunoprecipitation, purified ChIP DNA was analyzed in triplicates by qPCR, using SYBR Green master mix (Bio-Rad) and a CFX96 Real-Time System (Bio-Rad) with the settings described in Section 4.8. After each cycle, the amount of PCR product was measured (plate read), and at the end of each experiment, a melting curve was created (65.0 to 95 °C, plate read every 1.0 °C). Data analysis was realized according to the percent input method. For each primer pair, a standard curve of serially diluted DNA and a melting curve were performed, to ensure the efficiency, as well as the specificity, of PCR amplification and primer pairs. Error bars are representing the standard deviation of technical triplicates. In this study, the following primer pairs were used:

ctrl1TP:

F: 5′-GGCCCTTGGAAGAGAATGCT-3′

R: 5′-ACATCTGAGGACATTGCCCG-3′

MMP12_gb:

F: 5′-TTCTTGTCCCCTAGTCCAATGC-3′

R: 5′-GACGTGTGACTCTGGGCAA-3′

RBMS1_gb:

F: 5′-GTCAAGGTGGGAGAACTGCTTA-3′

R: 5′-CACACCACCACATGCAGTTAATT-3′

RPL11_2gb:

F: 5′-ACAGCTTTGGGTGATGCAGT-3′

R: 5′-TTGTTGGACCAAAACACGGC-3′

## Figures and Tables

**Figure 1 ijms-22-00678-f001:**
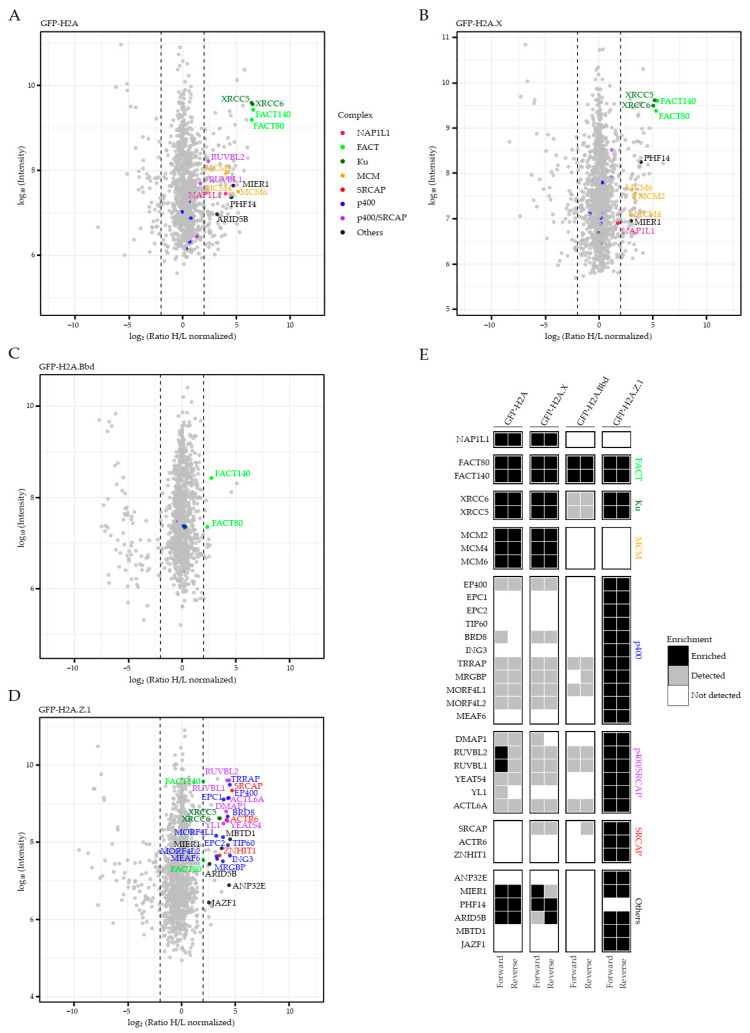
JAZF1 is member of H2A.Z chaperone complex(es). (**A–D**) Scatter plots of Green Fluorescent Protein (GFP) pull-downs for GFP–H2A (**A**), GFP–H2A.X (**B**), GFP–H2A.Bbd (**C)** and GFP–H2A.Z.1 (**D**)**.** HeLa Kyoto cells stably expressing GFP–H2A, –H2A.X, -H2A.Bbd or GFP–H2A.Z.1 were SILAC-labeled and subjected to single-step affinity purifications of soluble nuclear proteins, using GFP nanotrap beads. In each panel, the ratio of the identified proteins after MS is plotted. Nucleosome-assembly protein 1-like 1 (NAP1L1) is plotted in pink, members of the facilitates chromatin transcription (FACT) complex are depicted in bright green, members of the Ku complex in dark green, mini chromosome maintenance (MCM) proteins in orange, members of the SRCAP complex in red, of the p400 complex in blue and shared members of both complexes in purple. Other proteins are highlighted in black. Significantly enriched proteins of the abovementioned categories are labeled with names, non-significantly enriched but detected proteins are only highlighted with the respective color within the plot. Dotted lines represent selected ratio cutoff of 4 and 0.25. Shown is one out of two similar replicates. See Appendix A for a list of all identified proteins. (**E**) Heatmap of all identified chaperone proteins described in (**A**–**D**). Shown are enriched (black), detected but not enriched (gray) and not detected (white) proteins after forward or reverse SILAC–MS identifications. In forward and reverse experiments, selected ratio cutoff is 4 and 0.25, respectively. Shown is one of two similar biological replicates.

**Figure 2 ijms-22-00678-f002:**
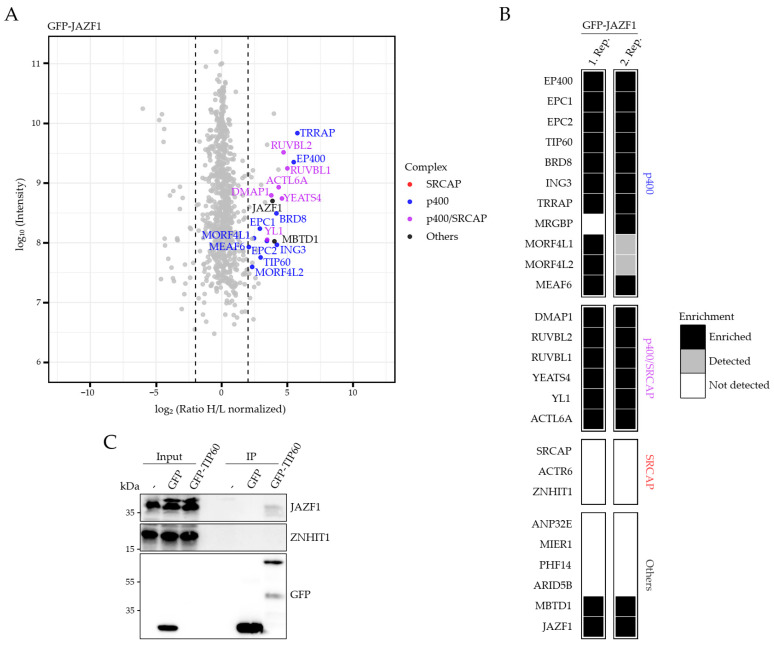
JAZF1 is a member of a TIP60- and MBTD1-containing p400 complex. (**A**) Scatter plot of GFP pull-downs for GFP–JAZF1. HeLa Kyoto cells transiently expressing GFP or GFP–JAZF1 were SILAC-labeled and subjected to single-step affinity purifications of soluble nuclear proteins, using GFP nanotrap beads. In each panel, the ratio of the identified proteins after MS is plotted. Members of the SRCAP complex are depicted in red, members of the p400 complex are in blue and shared members of both complexes are in purple. JAZF1 and MBTD1 are highlighted in black. Dotted lines represent selected ratio cutoff of 4. Shown is one of two biological replicates. See Appendix A for a list of all identified proteins. (**B**) Heatmap of p400, SRCAP and shared complex members identified in (**A**) and an additional replicate SILAC experiment. Shown are enriched (black), detected but not enriched (gray) and not detected (white) proteins after MS identifications. Selected ratio cutoff is 4. (**C**) Immunoblotting of JAZF1, SRCAP member ZNHIT1 and GFP upon transient transfection of HeLa Kyoto cells with GFP and GFP–TIP60, followed by GFP-trap pull-downs of soluble nuclear extracts.

**Figure 3 ijms-22-00678-f003:**
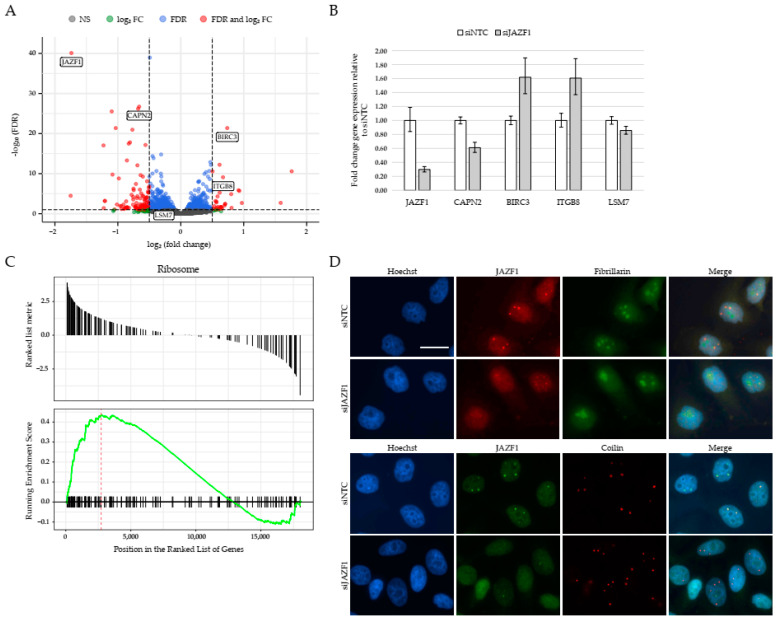
JAZF1 depletion leads to deregulation of genes involved in ribosome biogenesis. (**A**) Volcano-plot representation of differential expression analysis of genes upon JAZF1 depletion (siJAZF1 vs. siNTC), showing 162 statistically significant deregulated genes, 130 downregulated and 32 upregulated. Red dots represent statistically significant deregulated genes (log2FC > 0.5 or <−0.5 and False Discovery Rate (FDR) < 0.1); blue dots represent genes with significant statistical values (FDR < 0.1) that not reach our fold change threshold; green dots represent genes with significant expression changes (log2FC > 0.5 or <−0.5) without providing enough statistical evidence; gray dots represent genes with no statistical or fold-change significance. Genes that were validated by using RT-qPCR (**B**) are highlighted with boxes. See also Appendix A for PCA plot of deregulated genes upon JAZF1 knockdown. (**B**) RT-qPCR verification of two downregulated (JAZF1 and CAPN2) and two upregulated (BIRC3 and ITGB8) genes and one in its expression unchanged gene (LSM7) upon JAZF1 depletion. Shown is the fold change of three replicates normalized to HPRT1 expression. Error bars depict the SD of three replicates. Shown is one out of three biological replicates showing similar results. (**C**) Gene set enrichment analysis (GSEA) plot showing statistically significant and consistent differences of genes associated with the KEGG (Kyoto Encyclopedia of Genes and Genomes) pathway “Ribosome” upon JAFZ1 depletion. See also Appendix A for further GSEA analyses. (**D**) Immunofluorescence microscopy analysis of HeLa Kyoto cells co-stained with antibodies against JAZF1 (top, red; bottom, green), Fibrillarin (green, top) or Coilin (red, bottom) and Hoechst (DNA, blue) after control (siNTC) or JAZF1 (siJAZF1) knockdowns. Scale bar for all pictures = 20 µm.

**Figure 4 ijms-22-00678-f004:**
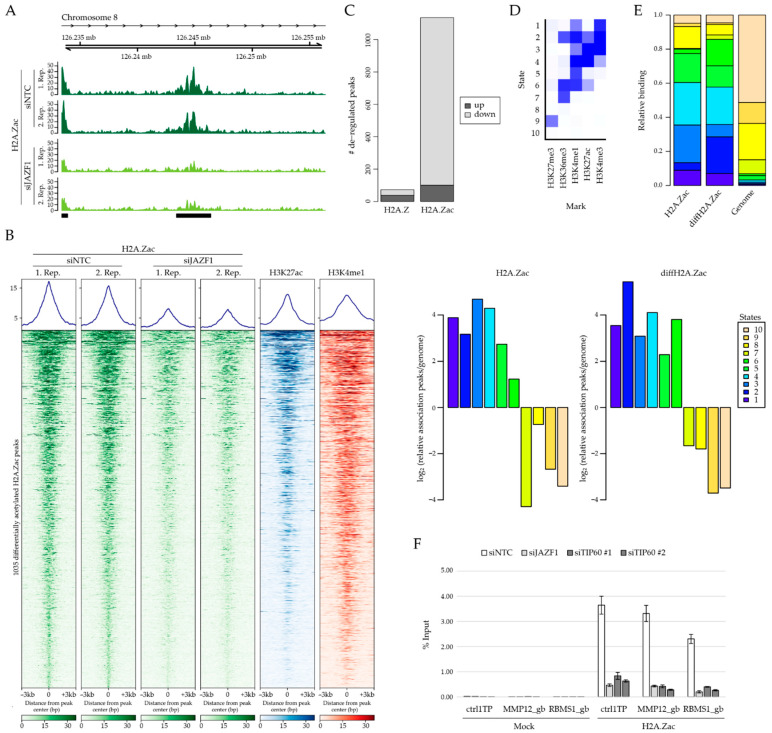
JAZF1 depletion leads to decrease in H2A.Z acetylation at regulatory regions. (**A**) Genome browser snap shot of human chromosome 8 locus as representative region displaying differential H2A.Zac ChIP-seq signals in two replicates of control (siNTC; non-target control siRNA; dark green) and JAZF1 knockdown (siJAZF1; light green). (**B**) ChIP-seq density heatmap of H2A.Zac upon control (siNTC) or JAZF1 (siJAZF1) knockdowns (two replicates, green), as well as visualization of the intensities of H3K27ac (blue, ENCODE) and H3K4me1 (red, ENCODE) at these differentially H2A.Z acetylated regions. Color intensities represents normalized and globally scaled tag counts. (**C**) Bar-plots visualizing deregulated H2A.Z or H2A.Zac peaks upon JAZF1-depletion. Notice that no major change is observed in H2A.Z peaks, but > 1000 H2A.Zac peaks are downregulated upon loss of JAZF1. (**D**) ChromHMM-based characterization of chromatin states [56,57]. The heatmap depicts the emission parameters of the HMM and describes the combinatorial occurrence of the individual histone modifications (mark) in different chromatin states (1–10). (**E**) Chromatin-state enrichment of global H2A.Zac and differentially deregulated H2A.Zac peaks (diffH2A.Zac) upon JAZF1 depletion in specific states (see (**D**)) calculated to frequency in complete genome (right). Notice enrichment of diffH2A.Zac peaks in states 2 and 6 that resemble H3K36me3-positive enhancer regions. See also Appendix A for further characterizations. (**F**) ChIP–qPCR verification of three diffH2A.Zac sites (gb: gene body) upon JAZF1 (light gray) and TIP60 depletion (two independent knockdowns: siTIP60 #1 and siTIP60 #2, dark gray, see also Appendix A for knockdown efficiency), compared to control knockdown (siNTC, white) using no (mock) or H2A.Zac antibodies. Shown is the respective enrichment as percentage of input signals. Error bars represent the SD of three replicates. Notice that TIP60 knockdown also leads to a similar reduction in H2A.Zac at these three diffH2A.Zac sites, compared to JAZF1 depletion.

## Data Availability

Raw and processed RNA-seq and the ChIP-seq data analyzed in this publication were transferred to NCBI’s Gene Expression Omnibus [90]. Data are accessible through the following GEO Series accession numbers: GSE163214 and GSE163318.

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
