# Peer review of "JAZF1, A Novel p400/TIP60/NuA4 Complex Member, Regulates H2A.Z Acetylation at Regulatory Regions"

_ijms, 2021, doi:10.3390/ijms22020678_

Round 1

Reviewer 1 Report

The manuscript of Procida and colleagues entitled “JAZF1, a novel p400/TIP60/NuA4 complex member, regulates H2A.Z acetylation at regulatory regions” is very interesting and very well written. This work is impressive and represents huge experiments well designed. I congratulate the authors. However, few minor points have to be improved before acceptance.

Minor points:

Figures 1 and 2 are blurred and not legible. Could the authors please upload the figures with higher resolution?

Could the authors explain their choice of cell line? Why did they use Hela cell line? Do they obtain similar results with another cancer cell line such as prostate cancer cells or endometrial cancer cells whose JAZ1 variants are known to increase the incidence of cancer?

In the case the results are specific to cervical cancers, it could be interesting to discuss the potential impact of these results for the clinical practice.

Reviewer 2 Report

The authors present a thorough quantitative proteomic analysis to identify interacting chaperones and/or remodeling complexes with human H2A variants, focusing on H2A.X, H2A.Bbd and H2A.Z. They discovered JAZF1 as a member of the p400 sub-complex and identify its impact on the levels of acetylation of H2A.Z and its role in the transcriptional regulation of ribosomal biogenesis.   The authors have prepared a concrete and precise study, with sound experimental methods and well-supported conclusions. Reading their manuscript was greatly interesting and satisfying. The only suggestion I have is to substantiate further their final conclusion. They nicely characterize the genomic distribution of H2A.Z acetylation at particular regulatory sites and speculate that TIP60 is the acetyltransferase responsible. It would be great to test whether the recruitment of TIP60 is affected by JAZF1 depletion. The authors clarify that ChIP-seq analysis of JAZF1 is not possible, but the TIP60 antibody has been used for genome-wide chromatin immunoprecipitations in many studies.   Minor comments   1. the resolution of the figures was very low and I could not read the protein names in any of the scatter plots or heat maps presented. 2. In Fig.2C there are 2 bands at the GFP-TIP60 IP lane blotted with GFP antibody and it is not clear what they are. 3. In Fig.3B there are no p-values of the RT-qPCR reactions.  
